# The adipocyte hormone leptin sets the emergence of hippocampal inhibition in mice

Camille Dumon[1†], Diabe Diabira[1], Ilona Chudotvorova[1], Francesca Bader[1,2], Semra Sahin[3], Jinwei Zhang[4], Christophe Porcher[1], Gary Wayman[3], Igor Medina[1], Jean-Luc Gaiarsa[1]*

[1]Aix-Marseille University UMR 1249, INSERM (Institut National de la Santé et de la Recherche Médicale) Unité 1249, INMED (Institut de Neurobiologie de la Méditerranée), Marseille, France; [2]Plateforme Post-Génomique, INMED, Marseille, France; [3]Program in Neuroscience, Department of Integrative Physiology and Neuroscience, Washington State University, Pullman, United States; [4]Institute of Biochemical and Clinical Sciences, Hatherly Laboratory, University of Exeter Medical School, Exeter, United Kingdom

**\*For correspondence:**
jean-luc.gaiarsa@inserm.fr

**Present address:** [†]Neurochlore Parc Scientifique et Technologique de Luminy, Bâtiment Beret Delaage, Zone Luminy Entreprises Biotech, Marseille, France

**Competing interests:** The authors declare that no competing interests exist.

**Abstract** Brain computations rely on a proper balance between excitation and inhibition which progressively emerges during postnatal development in rodent. γ-Aminobutyric acid (GABA) neurotransmission supports inhibition in the adult brain but excites immature rodent neurons. Alterations in the timing of the GABA switch contribute to neurological disorders, so unveiling the involved regulators may be a promising strategy for treatment. Here we show that the adipocyte hormone leptin sets the tempo for the emergence of GABAergic inhibition in the newborn rodent hippocampus. In the absence of leptin signaling, hippocampal neurons show an advanced emergence of GABAergic inhibition. Conversely, maternal obesity associated with hyperleptinemia delays the excitatory to inhibitory switch of GABA action in offspring. This study uncovers a developmental function of leptin that may be linked to the pathogenesis of neurological disorders and helps understanding how maternal environment can adversely impact offspring brain development.
DOI: https://doi.org/10.7554/eLife.36726.001

## Introduction

Most brain computations rely on a proper balance between excitation and inhibition, which progressively emerges during development. The γ-aminobutyric acid (GABA) is the main inhibitory transmitter in the adult brain. However, in rodents, at fetal and postnatal stages, GABA induces a membrane depolarization due to elevated intracellular chloride concentration ($[Cl^-]_i$) (*Ben-Ari et al., 2007*; *Sulis Sato et al., 2017*; *Owens and Kriegstein, 2002*). During the second postnatal week of life, the functional expression of the chloride extruder, $K^+$-$Cl^-$ type 2 co-transporter (KCC2), causes $[Cl^-]_i$ to decrease and consequently shifts the chloride-dependent GABAergic responses towards a more hyperpolarized value (*Rivera et al., 2015*). This developmental sequence is likely shifted toward fetal life in humans (*Chen and Kriegstein, 2015*; *Sedmak et al., 2016*). Rodent studies showed that defective chloride homeostasis plays a role in phenotypes associated with autism spectrum disorders (ASD) (*Tyzio et al., 2014*; *He et al., 2014*), Rett syndrome (*Banerjee et al., 2016*; *El-Khoury et al., 2014*), Down syndrome (*Deidda et al., 2015*) and Huntington's disease (*Dargaei et al., 2018*). Conversely, pharmacological manipulations aimed at restoring low $[Cl^-]_i$ improve neurological symptoms in rodents (*Tyzio et al., 2006*; *Deidda et al., 2015*; *Banerjee et al., 2016*; *Dargaei et al., 2018*) and

humans (*Lemonnier et al., 2017*). Therefore, unveiling the mechanisms controlling the schedule of the GABA developmental sequence is decisive to identify molecular targets to correct abnormal developmental trajectories.

Leptin is a 16 kDa cytokine produced and secreted mainly by the adipose tissue. In the adult, leptin regulates body weight and reproduction acting on specific hypothalamic nuclei (*Ahima and Flier, 2000*). However, rodent studies show that leptin can act beyond this classical role and may function as a neurotrophic signal (*Bouret, 2010*; *Harvey, 2013*; *Guimond et al., 2014*; *Briffa et al., 2015*). In both humans and rodents leptin levels surge during intense period of neuronal growth and synaptogenesis. Leptin levels remain elevated in children with early onset ASD (*Rodrigues et al., 2014*), Rett syndrome (*Blardi et al., 2009*) and Fragile X syndrome (*Lisik et al., 2016*). Likewise, maternal obesity which is associated with hyperleptinemia heightens the risk of ASD and other neuropsychiatric disorders in children (*Wang et al., 2016*; *Rivera et al., 1999*). These observations suggest that excess leptin early in life may play a role in neuropsychiatric disorders in humans.

Here, we examined how leptin impacts the GABAergic developmental sequence in the rodent hippocampus. We provide evidence that leptin controls the depolarizing-to-hyperpolarizing GABA switch during postnatal development and that maternal obesity associated with hyperleptinemia delays the GABA switch in offspring.

## Results

### Advanced onset of GABAergic inhibition in leptin-receptor deficient mice

We first compared the efficacy of GABA$_A$ receptor signaling in newborn long-form leptin-receptor (LepRb, the only leptin receptor able to activate intracellular pathway (*Ahima and Flier, 2000*) deficient (*db/db*) and wild type (wt) mice. To this aim we obtained non-invasive loose cell attached recordings of CA3 pyramidal neurons in acute hippocampal slices from *db/db* and wt littermates and investigated the effect of bath application of the GABA$_A$ receptor agonist isoguvacine (10 µM, 2 min) on their neuronal firing (*Figure 1A*). Consistent with the known depolarizing action of GABA in the newborn hippocampus (*Ben-Ari et al., 2007*), isoguvacine increased the firing of wt CA3 pyramidal neurons from P1 to P6 (*Figure 1B*). Remarkably, in *db/db* mice, the excitatory action of isoguvacine was only observed at P1 (*Figure 1B*). As early as P3, isoguvacine decreased the firing of the CA3 pyramidal *db/db* neurons (*Figure 1B*). At P15, the effect of isoguvacine on the firing of the CA3 pyramidal neurons was similar in wt and *db/db* mice (*Figure 1B*). Next to determine whether the depolarizing-to-hyperpolarizing GABA developmental sequence was affected, we investigated the reversal potential of evoked GABA$_A$ receptor-mediated postsynaptic currents (eGABA$_A$-PSCs) in *db/db* and wt littermates from P1 to P20 using gramicidin perforated voltage-clamp recording to prevent disruption of intracellular chloride concentration ([Cl$^-$]$_i$). GABA$_A$-PSCs were evoked in the presence of the glutamatergic receptor antagonists; NBQX (5 µM) and D-APV (40 µM) while voltage clamping the neurons at various potentials (*Figure 1C*). In wt CA3 neurons, the reversal potential of eGABA$_A$-PSCs (E$_{GABA}$) shifted from depolarizing to hyperpolarizing values during the second postnatal week of life (*Figure 1D*). In *db/db* neurons, this shift occurred earlier, few days after birth (*Figure 1D*, at P3 E$_{GABA}$ was $-38 \pm 8$ mV in wt and $-64 \pm 3$ mV in *db/db* neurons, p=0.023, two-tailed unpaired Student's *t-test*). In contrast the membrane potential at zero current was similar between wt and *db/db* neurons at all age investigated (*Figure 1D*). Altogether, these data show that the emergence of GABAergic inhibition is advanced in leptin-receptor deficient mice in vivo.

To further link leptin with the GABA sequence, we investigated the developmental profile of plasma leptin levels in wt mice. We found that circulating leptin levels were low at birth in wt, rose to a peak by the end of the first postnatal week and declined to low levels during the third postnatal week of life (*Figure 1E*). Interestingly, this developmental profile of leptin levels paralleled the developmental difference in E$_{GABA}$ observed between wt and *db/db* neurons (*Figure 1D,E*). Therefore, leptin levels surge during a critical developmental window in wt mice, and E$_{GABA}$ is impaired during this critical window in *db/db* mice. However, correlation does not mean causality. To address this point we used, newborn leptin-deficient (*ob/ob*) mice. We found that *ob/ob* mice also exhibit an advanced emergence of GABAergic inhibition in vivo, an effect partially restored by subcutaneous recombinant leptin injections to mimic the leptin surge occurring in wt mice in vivo (*Figure 1—figure*

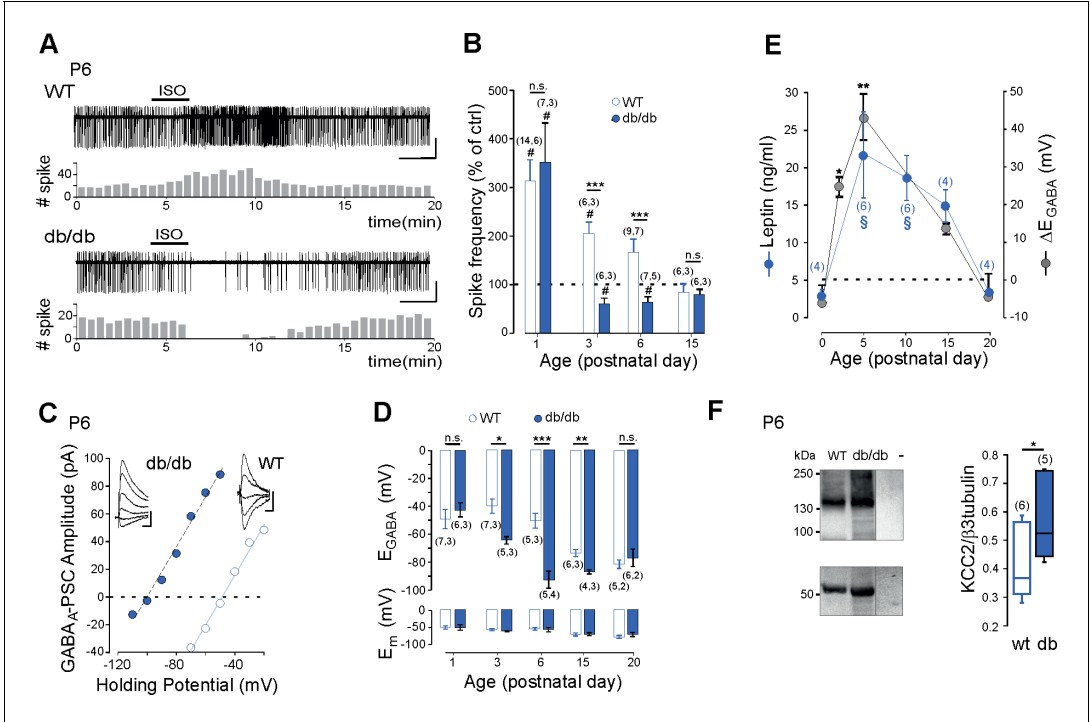

**Figure 1.** Early GABA developmental sequence in leptin-receptor deficient mice. (**A**) Cell attached recordings of CA3 pyramidal neurons on acute hippocampal slices. Scale bar, 2 min, 50 pA. Corresponding time course of spike frequency changes are shown under each trace. (**B**) Developmental changes of isoguvacine action on spike activity. Mean ± SEM. (**C**) Current-voltage relationships for evoked GABAergic synaptic currents. Insets: examples of GABAergic synaptic current evoked at holding potentials ranging from −110 to −60 mV (10 mV increment) in *db/db* and from −70 to −30 (10 mV increment) in wt CA3 pyramidal neuron. Scale bar, 10 ms, 20 pA. (**D**) Developmental changes in $E_{GABA}$ and Em at zero current. Mean ± SEM. In *B and* D, the number of cells recorded and number of mice used are indicated in parenthesis; [#]p<0.05 when compared to pre-isoguvacine values, two-tailed paired Student's *t*-test, *p<0.05, **p<0.01 and ***p<0.001 when compared to age-matched wt, two-tailed unpaired Student's *t*-test. (**E**) Developmental profile of plasma leptin levels in the wt (blue circle) and developmental profile of $\Delta E_{GABA}$ (gray circle). $\Delta E_{GABA}$ was calculated as the difference in $E_{GABA}$ values between the wt and *db/db* at each developmental stage depicted in *D*. Numbers in parenthesis indicate the number of mice used. Mean +SEM. [§]p<0.05 when compared to P0 plasma leptin values, *p<0.05 and **p<0.01 when compared to P0 $\Delta E_{GABA}$ values, one way ANOVA followed by a Tukey's *post hoc* test. (**F**) Left: representative immuno-blots for hippocampal panKCC2 and β3-tubulin in wt and *db/db* mice (first two lanes). The third lane (-) illustrates background (empty well). Right: box plots of normalized pan KCC2 in P6 wt and *db/db* hippocampi. Numbers in parenthesis indicate the number of mice used. *p<0.05, two-tailed unpaired Student's *t*-test.

DOI: https://doi.org/10.7554/eLife.36726.002

The following source data and figure supplements are available for figure 1:

**Source data 1.** Advanced onset of GABA inhibition in leptine receptor deficient mice.
DOI: https://doi.org/10.7554/eLife.36726.007

**Figure supplement 1.** Leptin controls GABA signaling in the mice hippocampus in vivo.
DOI: https://doi.org/10.7554/eLife.36726.003

**Figure supplement 1—source data 1.** Advanced onset of GABA inhibition in leptine deficient mice.
DOI: https://doi.org/10.7554/eLife.36726.004

**Figure supplement 2.** KCC2 expression in *db/db* hippocampal neurons in vivo and leptin-treated hippocampal neuronal cultures in vitro.
DOI: https://doi.org/10.7554/eLife.36726.005

**Figure supplement 3.** Raw blots for panel F (WT and db).
DOI: https://doi.org/10.7554/eLife.36726.006

*supplement 1*). Altogether, these data show that leptin surge controls chloride homeostasis and the emergence of functional GABAergic inhibition in vivo.

The developmental switch in GABA polarity is mainly due to the functional expression of KCC2 (*Medina et al., 2014*; *Rivera et al., 1999*). We therefore assessed mRNA expression of this Cl⁻ co-transporter in wt and *db/db* mice at P6 using quantitative qRT-PCR from isolated hippocampi. KCC2 mRNA levels were enhanced in *db/db* hippocampi compared to wt (from 1.6 ± 0.1 in wt to 2.6 ± 0.3

in *db/db* mice, n = 6 for both, p=0.038, two-tailed unpaired Student's *t*-test, not shown). In contrast, NKCC1 mRNA levels were not altered (1.1 ± 0.2 and 1.2 ± 0.2 in respectively wt and *db/db* mice, n = 6 for both, p=0.7, two-tailed unpaired Student's *t*-test, not shown). To determine whether increased mRNA levels results in increased expression of KCC2, we assessed KCC2 protein levels in P6 hippocampi by western blotting. KCC2 protein expression was up-regulated in *db/db* hippocampi compared to wt (+33 ± 6%, n = 6 wt and 5 db/db hippocampi, p=0.02 two-tailed unpaired Student's *t*-test, Figure 3F). Immuno-labeling confirmed that KCC2 protein expression was up-regulated in *db/db* hippocampi at P6 compared to wt (*Figure 1—figure supplement 2*). KCC2 expression was also up-regulated in *ob/ob* hippocampi compared to wt (from 0.38 ± 0.07 to 0.97 ± 0.24 when normalized to β3 tubulin, n = 6 for both, p=0.04, two-tailed unpaired Student's *t*-test, not shown). NKCC1 expression was however similar between wt and *db/db* neurons (the NKCC1/β3 tubulin ratio was 0.56 ± 0.01 and 0.55 ± 0.02 in respectively wt (n = 3) and *db/db* (n = 3) mice, p=0.8 two-tailed unpaired Student's *t*-test, Supplementary file 1). Thus a deficit in leptin signaling leads to an earlier expression of KCC2 and an advanced GABA switch to inhibition in vivo.

## Leptin controls chloride homeostasis in vitro

We next asked whether leptin directly acts on hippocampal cells to control Cl$^-$ homeostasis. To address this point, we first evaluated the effect of a transient exposure to leptin (100 nM, 24 hr) on Cl$^-$ homeostasis on immature hippocampal cultures (6 DIV) when KCC2 activity is low, and in more mature cultures (15 DIV) when KCC2 activity is high and GABA had switched to hyperpolarized values (*Friedel et al., 2015*). Using gramicidin perforated voltage-clamp recordings, we assessed the reversal potential of GABA$_A$ receptor-mediated currents (E$_{GABA}$) induced by brief focal applications of isoguvacine in the presence of the NKCC1 blocker bumetanide (10 µM), to exclude cross regulation between the 2 co-transporters. We found that leptin treatment had no effect on E$_{GABA}$ in immature neurons (E$_{GABA}$ = −61 ± 1 mV in control (19 neurons) and −61 ± 2 mV in leptin-treated (19 neurons) cultures, 3 cultures for both, *Figure 2B*), but induced a depolarizing shift of isoguvacine currents from −99 ± 2 mV in control (21 neurons) to −85 ± 2 mV in leptin-treated (13 neurons) cultures (6 cultures for both, p=0.001, ANOVA followed by a Tukey's *post hoc* test, *Figure 2A,B*). The leptin-induced shift was abolished in neurons transfected with validated specific sh-RNAs (*Dhar et al., 2014*) to silence the expression of the LepRb (*Figure 2B*). In neurons transfected with a scramble shRNA, leptin was still able to depolarize E$_{GABA}$ (*Figure 2B*). Treatment with the selective KCC2 blocker VUO463271 (10 µM) shifted E$_{GABA}$ towards more depolarized values and occluded the effect of leptin (*Figure 2B*). Leptin also decreased the rate of Cl$^-$ extrusion by KCC2 in cultured neurons preloaded with Cl$^-$, an effect mimicked by VUO463271 (*Figure 2C,D*). The half-recovery time was 194 ± 17 s, 563 ± 143 s and 625 ± 176 s in respectively control (n = 7 neurons), leptin-treated (6 neurons, p=0.018 when compared to control, two-tailed unpaired Student's *t*-test) and VUO463271-treated (n = 4 neurons, p=0.79 when compared to leptin-treated, two-tailed unpaired Student's *t*-test, Supplementary file 1) cultures. These data therefore show that leptin acts on hippocampal leptin receptor to down regulate KCC2 activity shifting E$_{GABA}$ towards depolarizing values in hippocampal neuronal cultures.

We next asked whether leptin treatment altered KCC2 and NKCC1 expression. Western blotting revealed that the expression of the two chloride co-transporters was decreased in leptin-treated cultures (−85 ± 0.8 and −66 ± 0.6% respectively, 5 cultures for both, p=2.01E-6 and 0.0008 respectively, two-tailed unpaired Student's *t*-test, *Figure 3D*). However, the ratio of the KCC2/NKCC1 protein was reduced following leptin treatment (from 1.05 ± 0.19 to 0.49 ± 0.07, p=0.02, two-tailed unpaired Student's *t*-test, *Figure 3D*). Immuno-labeling confirmed that KCC2 protein was decreased in leptin-treated cultures (*Figure 1—figure supplement 2*).

The total amount of KCC2 protein is not an accurate indicator of its activity since, to extrude Cl$^-$, KCC2 has to localize at the plasma membrane (*Medina et al., 2014*). We therefore assessed the abundance of KCC2 expressed in different cell compartments of control and leptin-treated cultured hippocampal neurons, using a KCC2 construct tagged in an external loop with a fluorescent protein (KCC2-pH$_{ext}$) (*Friedel et al., 2015*). Combined with a multistep immuno-labeling protocol, this construct allows to visualize the total amount of KCC2-pH$_{ext}$ expressed by neurons (F$_t$), the amount of KCC2-pH$_{ext}$ present at the cell surface (F$_m$) and the amount of KCC2-pH$_{ext}$ internalized (F$_i$) (*Figure 3A*). F$_t$ was similar in control and leptin-treated cultured neurons (1.1 ± 0.13 vs 1.3 ± 0.18 a. u., 16 and 19 neurons respectively, 3 cultures, p=0.89, ANOVA followed by a Tukey's *post hoc* test,

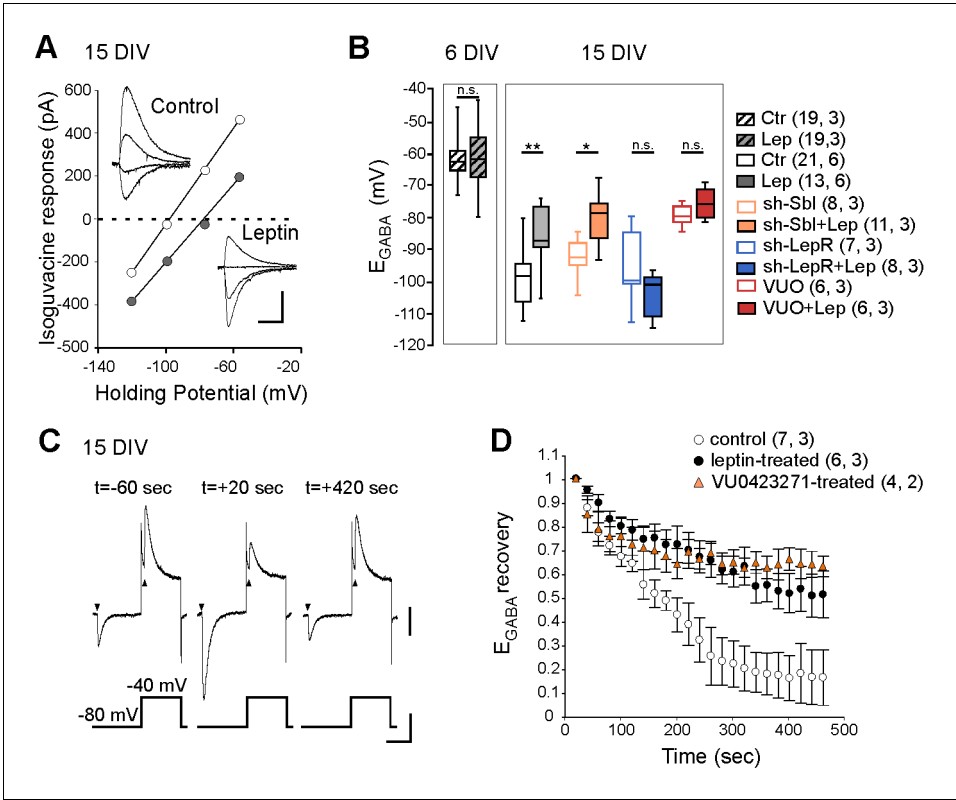

**Figure 2.** Leptin down-regulates KCC2 activity in cultured hippocampal neurons. (**A**) I-V relationships for isoguvacine currents in vehicle (control) and leptin-treated (100 nM, 24 hr) hippocampal (15 DIV) cultures. Gramicidin perforated patch clamp recordings. Insets depict the isoguvacine currents. Scale bar, 500 ms, 100 pA. (**B**) Box plots of $E_{GABA}$ in the indicated conditions. *p<0.05, **p<0.01, ANOVA followed by a Tukey's *post hoc* test. (**C**) Examples of isoguvacine currents (arrow heads) recorded at −80 and −40 mV before (t=-60 sec) and after (t = +20 and +420 s) neuronal chloride loading in control neuronal culture (15 DIV). Gramicidin perforated patch clamp recordings. Scale bar, 100 pA, 40 mV, 1 s. (**D**) Summary plots of normalized $E_{GABA}$ recovery after neuronal chloride loading in the indicated conditions. Mean ± SEM. In *B* and *D*, the number of cells recorded and number of cultures used are indicated in parenthesis.

DOI: https://doi.org/10.7554/eLife.36726.008

The following source data is available for figure 2:

**Source data 1.** Leptin decreases KCC2 activity in cultured hippocampal neurons.
DOI: https://doi.org/10.7554/eLife.36726.009

not shown). However, $F_m$ was lower (1 ± 0.16 vs 0.44 ± 0.09 a.u., p=0.0028, ANOVA followed by a Tukey's *post hoc* test) and $F_i$ was higher (0.9 ± 0.18 vs 1.7 ± 0.15 a.u., p=0.04, ANOVA followed by a Tukey's *post hoc* test) in leptin-treated cultures compared to control (*Figure 3A,B*). Both effects were abolished when the expression of LepRb was silenced with two different specific sh-RNAs (*Dhar et al., 2014*) (*Figure 3B*). One batch of cultures was routinely transfected with KCC2 construct tagged in the intracellular N-terminus with a fluorescent protein as control experiment for cell membrane integrity during live-cell immuno-labelling, (KCC2-pH$_{int}$, *Figure 3A*). The multistep immuno-labeling protocol didn't detect membrane expressed or internalized KCC2-pH$_{int}$ (*Figure 3B*). Altogether these data show that leptin reduces the expression of KCC2 and its plasma membrane stability in cultured hippocampal neurons.

The membrane expression and transport activity of KCC2 strongly depend on the phosphorylated state of its intracellular C-terminus domain (*Medina et al., 2014*). Phosphorylation on the threonine 906 and 1007 residues (T906, T1007) inhibits KCC2 activity and enhances KCC2 endocytosis, while phosphorylation of the serine 940 residue (S940) enhances KCC2 activity (*Medina et al., 2014*). We therefore assessed the phosphorylated state of KCC2 following leptin treatment in neuronal cultures.

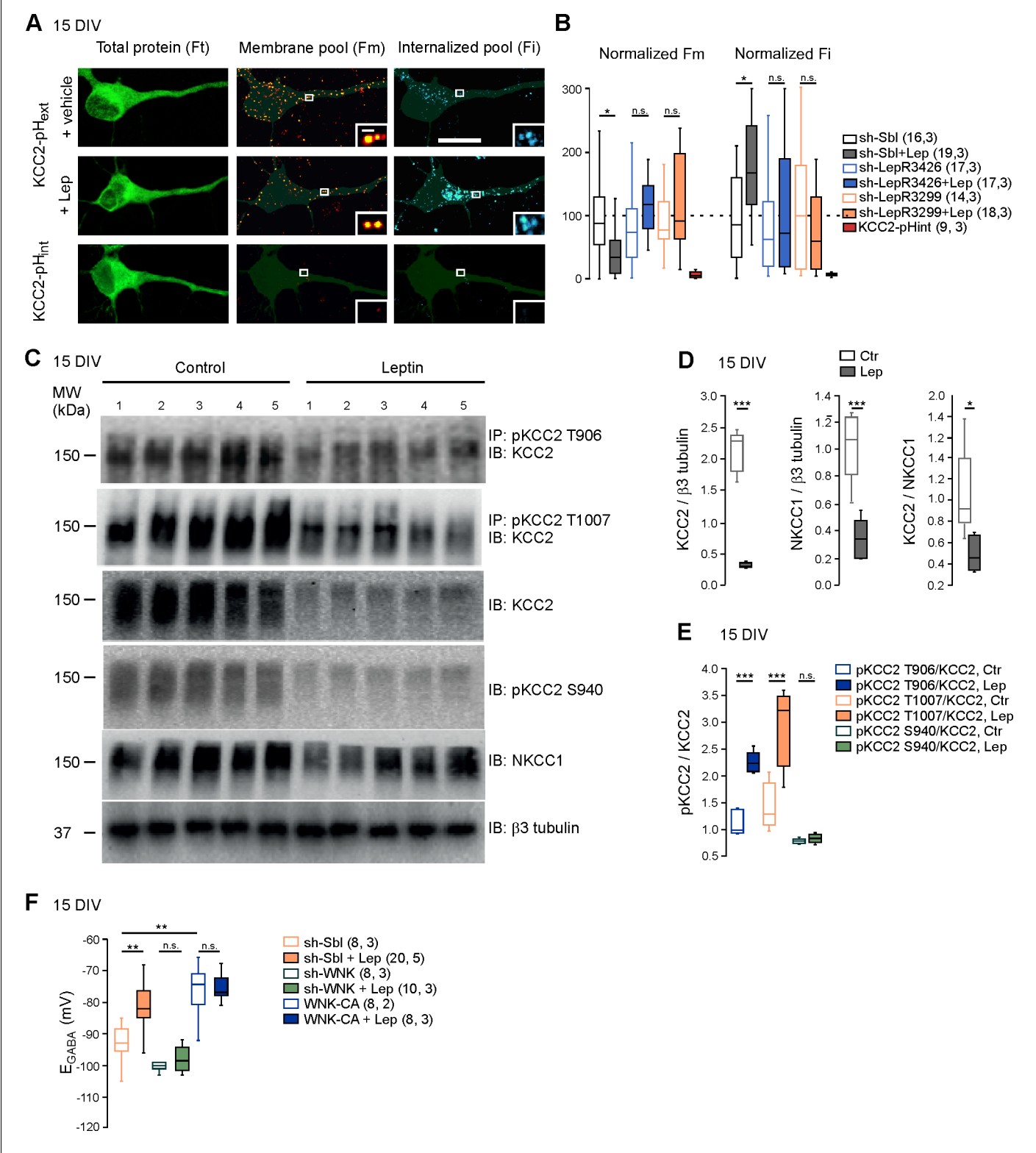

**Figure 3.** Leptin reduces the plasma membrane expression of KCC2 and modulates its phosphorylation state in cultured hippocampal neurons. (A) Representative images illustrating total, membrane and internalized pools of KCC2 with external tag (KCC2-pHext) in vehicle and leptin-treated (100 mM, 24 hr) cultured hippocampal neurons expressing a scramble Sh-RNA (Sh-Sbl). Neurons expressing KCC2 with internal tag (KCC2-pHint) were proceeded in parallel experiments to ensure that immunocytochemistry on living neurons does not permeabilized the membrane. Scale bars 20 µm and 1 µm. (B) Box plots of normalized membrane (Fm) and internalized (Fi) fluorescence in vehicle and leptin-treated (+Lep) cultured neurons expressing the

*Figure 3 continued on next page*

*Figure 3 continued*

indicated constructs. *p<0.05, one way ANOVA followed by a Tukey's *post hoc* test. (C) Western blots and quantifications (D and E) of KCC2, NKCC1, KCC2/NKCC1 ratio and the threonine 906, threonine 1007 and serine 940-phosphorylated forms of KCC2 in control and leptin (100 nM, 24 hr)-treated hippocampal neuronal cultures (DIV15, five independent neuronal cultures). ***p<0.001, two-tailed unpaired Student's *t-test*. (F) Box plots of $E_{GABA}$ in the indicated conditions. Gramicidin perforated patch clamp recordings were performed on hippocampal neuronal cultures at 15 DIV. **p<0.01, ***p<0.001, one way ANOVA followed by a Tukey's *post hoc* test. In B and F, the number of cells recorded and number of cultures used are indicated in parenthesis.

DOI: https://doi.org/10.7554/eLife.36726.010

The following source data is available for figure 3:

**Source data 1.** Leptine controls the membrane expression of KCC2 and its phosphorylated state in hippocampal culture via the WNK pathway.

DOI: https://doi.org/10.7554/eLife.36726.011

We found that the total amount of phospho-T906 KCC2 (pKCC2 T906), phospho-T1007 KCC2 (pKCC21007) and phospho-S940 KCC2 (pKCC2S940) protein were decreased in leptin-treated neuronal cultures (*Figure 3C*). However, the ratio of pKCC2 T906 to total KCC2 protein (pKCC2T906/KCC2) and pKCC2 T1007 to total KCC2 protein (pKCC2T1007/KCC2) were increased hence showing that the relative expression of these phosphorylated forms of KCC2 were increased by leptin treatment: the pKCC2T906/KCC2 ratio increased from $1.1 \pm 0.1$ to $2.2 \pm 0.08$, and the pKCC2T1007/KCC2 ratio increased from $1.4 \pm 0.2$ to $2.9 \pm 0.3$ (p=0.0003 and p=0.004, respectively, two-tailed unpaired Student's *t*-test, 5 independent neuronal cultures, *Figure 3C,E*). The relative expression of phospho-S940, calculated as the ratio of pKCC2 S940 to total KKC2 protein (pKCC2S940/KCC2) was however not modified (the ratio of pKCC2S940/KCC2 was $0.77 \pm 0.02$ in control and $0.083 \pm 0.03$ in leptin-treated cultures, p=0.26, two-tailed unpaired Student's *t*-test, 5 independent neuronal cultures, *Figure 3C,E*). These results show that leptin treatment alters the phosphorylated state of KCC2 in hippocampal neuronal cultures.

WNK1 activity is required for KCC2 T906 and T1007 phosphorylation (*Inoue et al., 2012*; *Friedel et al., 2015*). To test whether WNK1 is involved in the leptin-induced regulation of chloride homeostasis, we expressed previously validated specific sh-RNA (*Friedel et al., 2015*) to silence WNK1 expression (sh-WNK) or constitutively active WNK1 mutant (WNK-CA) in hippocampal neuronal cultures and measured $E_{GABA}$ from gramicidin-perforated patch-clamp recordings after 24 hr of vehicle or leptin (100 nM) treatment (*Figure 3F*). In scramble expressing neurons, leptin treatment led to a depolarizing shift of $E_{GABA}$ from $-93 \pm 2$ mV to $-80 \pm 3$ mV, n = 8 and 20 neurons respectively, p=0.001, one way ANOVA followed by a Tukey's *post hoc* test, *Figure 3F*). Genetic silencing of WNK1 prevented the depolarizing shift of $E_{GABA}$ induced by leptin-treatment (p=0.89 compared to control sh-WNK expressing neurons, one way ANOVA followed by a Tukey's *post hoc* test, *Figure 3F*). In contrast, expressing WNK1-CA produced a depolarizing shift of $E_{GABA}$ ($E_{GABA}$=-75 $\pm$ 3.5 mV, p=0.001, one-way ANOVA followed by a Tukey's *post hoc* test when compared to scramble expressing neurons) and occluded the leptin action (p=0.89, one way ANOVA followed by a Tukey's *post hoc* test, *Figure 3F*). Altogether these data suggest that leptin modulates the activity of KCC2 via a WNK1-dependent pathway in cultured hippocampal neurons.

## Maternal obesity and neonatal hyperleptinemia delay the emergence of GABAergic inhibition

Maternal obesity causes excess of leptin in offspring (*Valleau and Sullivan, 2014*; *Tessier et al., 2013*). We therefore asked whether maternal obesity may affect the GABA developmental sequence in the offspring. Female mice were fed with normal (ND) or high fat (HFD) diet (*Figure 4—figure supplement 1*). After 6 weeks, females were mated and maintained under their respective diet during the gestation and lactation period. Pups of HFD-induced obese dams (DIO-pups) showed higher levels of serum leptin compared to offspring of ND dams (ND-pups) ($16 \pm 1$ ng/ml (n = 6) vs $32 \pm 4$ ng/ml (n = 4) in respectively P10-P15 ND-pups and DIO-pups, p=0.0049, two-tailed unpaired Student's *t*-test, not shown). We investigated the effect of bath applied isoguvacine on the firing of CA3 hippocampal slices. We found that the excitatory-to-inhibitory switch of isoguvacine actions was delayed by about one week in DIO-pups compared to ND-pups (*Figure 4A,B*). Thus, isoguvacine increased the neuronal firing in ND-pups at P5 and decreased it starting from P10. In HFD-

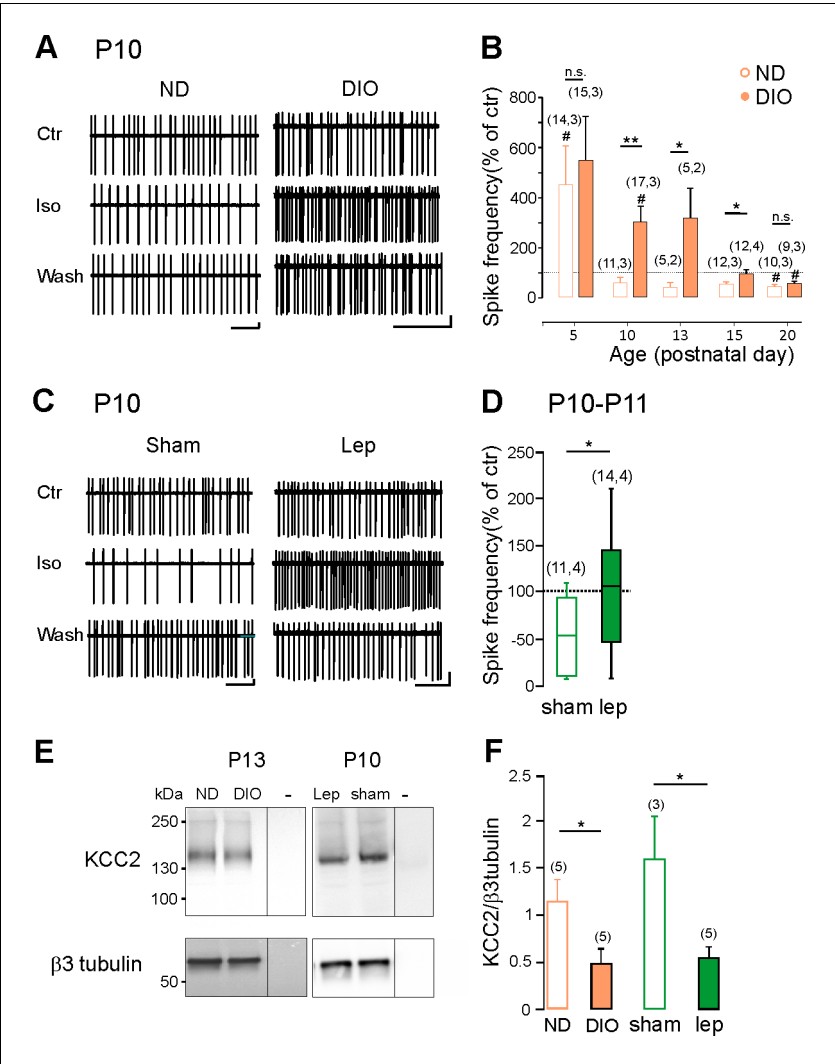

**Figure 4.** Hyperleptinemia and maternal obesity delay the GABA developmental sequence and downregulate KCC2 expression. (A) Cell attached recordings of CA3 pyramidal neurons on acute hippocampal slices obtained from pups of normal diet (ND) and diet-induced obese (DIO) dams at P10. (B) Developmental changes of isoguvacine action on spike frequency. Mean +SEM. (C) Cell attached recordings of CA3 pyramidal neurons on acute hippocampal slices obtained from vehicle-treated (sham) and leptin-treated mice at P10. (D) Box plots of isoguvacine action on spike activity. In *B* and *D,* number of cells recorded and number of mice used are indicated in parenthesis; #p<0.05 when compared to pre-isoguvacine values, two-tailed paired Student's *t*-test and *p<0.05 and **p<0.01 when compared to age matched ND-pups (B and C) or sham-pups (E), two-tailed unpaired Student's *t*-test. (E) Representative immuno-blots for hippocampal panKCC2 and β3-tubulin in offspring of DIO and ND dams at P13 and in control (sham) and leptin-treated (Lep) mice at P10. The third lanes (-) illustrate background (empty wells). (F) Normalized panKCC2 immunoreactivity in ND (n = 6 pups) and sham (n = 3 pups), in offspring of DIO (n = 5 pups) and in leptin-treated mice (n = 5 pups). Mean +SEM. *p<0.05, two-tailed unpaired Student's *t*-test.

DOI: https://doi.org/10.7554/eLife.36726.012

The following source data and figure supplements are available for figure 4:

**Source data 1.** Maternal obesity and hyperleptinemia delayed the emergence of functional GABAergic inhibition.
DOI: https://doi.org/10.7554/eLife.36726.016

**Figure supplement 1.** High fat diet induced obesity.
DOI: https://doi.org/10.7554/eLife.36726.013

**Figure supplement 2.** Raw blots for panel E (ND and DIO).
DOI: https://doi.org/10.7554/eLife.36726.014

**Figure supplement 3.** Raw blots for panel E (Lep and sham).

*Figure 4 continued on next page*

*Figure 4 continued*

DOI: https://doi.org/10.7554/eLife.36726.015

pups, isoguvacine increased the firing frequency up to P13. To assess whether hyperleptinemia mimicked the effect of maternal obesity, wt pups were treated with sub-cutaneous injections of recombinant leptin (5 mg/kg, twice a day) from P2 to P10. At P10-11, isoguvacine decreased the neuronal firing of CA3 pyramidal neurons in vehicle-treated mice but not in leptin-treated mice (*Figure 4C,D*).

Further, western blot analysis revealed that the expression of KCC2 was decreased in DIO-pups compared to ND-pups ($-51 \pm 12\%$, p=0.03, two-tailed unpaired Student's $t$-test, *Figure 4E,F*) and in leptin-treated mice compared to vehicle-treated mice ($-65 \pm 7\%$, p=0.03, two-tailed unpaired Student's $t$-test, *Figure 4E,F*). Thus, hyperleptinemia as maternal obesity delays the GABA developmental sequence and down regulates KCC2 expression in offspring.

## Discussion

Our data demonstrate that leptin is a key factor in setting the direction of $GABA_A$ receptor-mediated signaling in the developing hippocampus. We show that leptin exposure of hippocampal cultures leads to a depolarizing shift of $GABA_A$ receptor-mediated responses and down-regulates KCC2 expression and membrane stability. Furthermore, in the absence of leptin signaling, hippocampal neurons show an early expression of KCC2 and an advanced development of GABAergic inhibition in vivo. Conversely, excess leptin down regulates the expression of KCC2 and delays the emergence of GABAergic inhibition in vivo. Previous studies have reported that the GABA switch is delayed in mice lacking oxytocin (*Tyzio et al., 2014*; *Leonzino et al., 2016*) or the thyroid hormones (*Friauf et al., 2008*; *Sawano et al., 2013*). Thus, the functional maturation of GABAergic synapses depends on an optimal and timely balance level of hormones that accelerate (oxytocin, thyroid hormones) or retard (leptin) the GABA switch.

We have shown that leptin regulates the activity of KCC2 at the transcriptional and post-translational levels in vivo. The downstream pathway linking the long-form leptin receptor b (LepRb) and the functional expression of KCC2 remains to be elucidated. However, our data suggest that early in development, leptin surge promotes the phosphorylation of the threonine 906 and 1007 (T906/T1007) residues of KCC2 and reduces its membrane expression and activity via a WNK1-dependent pathway. This pathway seems to be quite specific in mediating leptin action, as we did not detect any changes in the phosphorylation of Ser940, a residue phosphorylated/dephosphorylated under other circumstances via protein kinase C (*Lee et al., 2011*). As development progresses, when leptin levels return to low level, the WNK1-dependent inhibitory action of KCC2 activity is removed allowing the depolarizing-to-hyperpolarizing transition of GABA actions in the developing rodent hippocampus. Our model is supported by several lines of evidence obtained in vitro. We have shown that genetic silencing or activation of WNK1, respectively, prevents or occludes the effect of leptin on $E_{GABA}$. Moreover, we have shown that leptin treatment does not affect $E_{GABA}$ in immature cultured neurons, at a developmental stage when the activity of KCC2 is low and the activity of the WNK-pathway is high (*Friedel et al., 2015*). Finally, we have shown that leptin enhances the phosphorylation of the T906/T1007 residues of KCC2, promotes the endocytosis of KCC2 and decreases its membrane expression. Of note, WNK1-regulated de-phosphorylation of the T906/1007 residues of KCC2 contributes to the depolarizing-to-hyperpolarizing GABA sequence in the developing hippocampus (*Inoue et al., 2012*; *Friedel et al., 2015*). Alanine substitutions of the T906/T1007 residues within the C-terminus of KCC2, mimicking a dephosphorylated state of KCC2, yield to hyperpolarized reversal potentials of GABA in neuronal cultures, ranging between $-90$ to $-100$ mV (*Titz et al., 2015*), that is the same order of magnitude to the values observed in *db/db* neurons (*Figure 1D*). Thus, the lack of leptin signaling in *db/db* and *ob/ob* mice may prevent the developmental WNK1-regulated changes in T906/T1007 KCC2 phosphorylation in vivo, thereby leading to an advanced emergence of GABA inhibition in the newborn rodent hippocampus.

Rodent studies show that the ability of GABA to depolarize immature neurons plays an essential role in the assembly of functional networks during development (*Ben-Ari et al., 2012*). Advanced GABAergic inhibition or premature expression of KCC2 lead to long lasting disturbance of glutamatergic and GABAergic inputs and behavioral abnormalities in adult mice (*Wang and Kriegstein,*

*2008*; *Wang and Kriegstein, 2011*; *Deidda et al., 2015*; *Cancedda et al., 2007*; *Chudotvorova et al., 2005*). Accordingly, hippocampal network formation and function are impaired in *db/db* mice (*Guimond et al., 2014*; *Harvey, 2013*; *Van Doorn et al., 2017*). However, it is impossible to ascertain whether the early emergence of GABA inhibition is a direct cause of these abnormal behaviors since leptin also directly impacts many aspect of brain development (*Guimond et al., 2014*; *Harvey, 2013*; *Bouret, 2010*) that may contribute to behavioral deficit in *db/db* mice.

Although translating animal research to the human situation is difficult, the GABA developmental sequence and leptin surge occurring during the second postnatal week of life in mouse are shifted towards fetal stage in humans (*Chen and Kriegstein, 2015*; *Sedmak et al., 2016*; *Valleau and Sullivan, 2014*). This seems in contradiction with the classical satiety function of leptin since during pregnancy nutriments availability should be enhanced to optimize fetal growth. However, animal studies suggest the existence of a pregnancy-induced leptin resistance of the mother's brain and accumulating evidence indicate that leptin has numerous actions before birth on fetal growth (*Briffa et al., 2015*). Importantly, leptin deficiency in this period is associated with long term consequences such as cognitive defects, anxiety, and depression. Conversely, leptin replacement alleviates these symptoms, suggesting that leptin is a crucial factor in brain development and mental health (*Wang et al., 2016*; *Rivera et al., 2015*). Thus, during pregnancy, leptin may act beyond its classical satiety role and operates as an important developmental signal of fetal brain development.

Our data also indicate that excess leptin delays the GABA switch in offspring. Defective chloride homeostasis primes the brain to malfunction contributing to autistic-like phenotypes associated with neurological disorders (*Tyzio et al., 2014*; *He et al., 2014*; *Deidda et al., 2015*; *Banerjee et al., 2016*). Moreover, it is noticeable that bumetanide treatment, that restores low $[Cl^-]_I$ and alleviates autistic-like symptoms in rodents (*Tyzio et al., 2014*; *Banerjee et al., 2016*), also improves the symptoms of ASD in humans (*Lemonnier et al., 2017*). The ability of leptin to regulate chloride homeostasis may therefore have important implications not only in health, but also in the emergence of neurological disorders associated with abnormal GABAergic transmission. ASD are often associated with higher levels of leptin in circulation (*Rodrigues et al., 2014*; *Rodrigues et al., 2014*; *Lisik et al., 2016*). Interestingly, maternal obesity which is associated with hyperleptinemia (*Valleau and Sullivan, 2014*; *Tessier et al., 2013*) increases the risk of developing mental and behavioral disorders in offspring (*Wang et al., 2016*; *Rivera et al., 2015*), such as attention deficit hyperactivity disorder (ADHD), ASD, anxiety, depression, schizophrenia and impairments in cognition, which among many modifications, are associated with altered GABAergic transmission. Animal models of maternal high-fat diet-induced obesity also document persistent changes in offspring behavior such as hyperactivity, impairments in social behavior, depressive-like behaviors and diminished cognition (*Rivera et al., 2015*).

In conclusion, optimal levels of leptin are critical during development for a timely emergence of GABAergic inhibition. Epidemiological studies showed that the prevalence of obesity among child-bearing age women has steadily increased during the past 20 years in most developed nations (*An and Xiang, 2016*; *Flegal et al., 2012*). Leptin thus lies at the crossroad between neurological, metabolic and nutritional disorders that may arise from societal changes in lifestyle, stress level and eating behaviors.

# Materials and methods

**Key resources table**

| Reagent type (species) or resource | Designation | Source or reference | Identifiers | Additional information |
|---|---|---|---|---|
| Genetic reagent (M. *musculus*) | B6.Cg-Lepob/J | The Jackson Laboratory | RRID:IMSR_JAX:000632 | |
| Genetic reagent (M. *musculus*) | B6.BKS-Leprdb | The Jackson Laboratory | RRID:IMSR_JAX:000697 | |

*Continued on next page*

*Continued*

| Reagent type (species) or resource | Designation | Source or reference | Identifiers | Additional information |
|---|---|---|---|---|
| Transfected DNA construct | shLepR 3426 | PMID:24877561, DOI: 10.1210/me.2013–1332 | | Dr. G. Wayman (Washington State University) |
| Transfected DNA construct | shLepR 3299 | PMID:24877561, DOI: 10.1210/me.2013–1332 | | Dr. G. Wayman (Washington State University) |
| Transfected DNA construct | WNK shRNA | PMID: 26126716, DOI: 10.1126 /scisignal.aaa0354 | | |
| Transfected DNA construct | WNK-CA | PMID: 26126716, DOI: 10.1126 /scisignal.aaa0354 | | |
| Transfected DNA construct | KCC2-pHluorin | PMID: 24928908, DOI: 10.15252 /embr.201438840 | | |
| Antibody | Mouse anti-β3 tubulin | Sigma-Aldrich | RRID:AB_477590, Cat# T8660 | WB (1:10 000) |
| Antibody | Rabbit anti-KCC2 | US Biological | RRID:AB_2188802, Cat# K0120-07 | WB (1:2000) |
| Antibody | Donkey anti-chiken Alexa488 | Fluoprobes | RRID: AB_2686906, Cat# FP-SA5110 | IHC (1:1000) |
| Antibody | Chicken anti-MAP2 | Abcam | RRID:AB_2138153, Cat# ab5392 | IHC (1:2000) |
| Antibody | Donkey Anti-rabbit Cy3 | Chemicon | RRID:AB_92588, Cat# AP182C | IHC (1:1000) |
| Antibody | mouse Anti-GFP | Novus Biologicals | RRID:AB_531011, Cat# NB 600–597 | |
| Antibody | NKCC1 total | the Division of Signal Transduction Therapy Unit (DSTT) at the University of Dundee | dundee (S022D) | WB (1 mg/ml) |
| Antibody | pan KCC2 | | | Dr. C. Rivera (University of Helsinki) |
| Antibody | KCC2 Ser940 | Novus Biologicals | Cat# NBP2-29513 | WB (1 mg/ml) |
| Antibody | KCC2a phosphoT1007 | the Division of Signal Transduction Therapy Unit (DSTT) at the University of Dundee | dundee (S959C) | WB (1 mg/ml) |
| Antibody | KCC2a phosphoT906 | the Division of Signal Transduction Therapy Unit(DSTT) at the University of Dundee | dundee (S959C) | WB (1 mg/ml) |
| Oligonucleotides | Slc12a2 (NKCC1) | Qiagen | QT00197785 | |
| Oligonucleotides | Slc 12a5 (KCC2) | Qiagen | QT00145327 | |
| Oligonucleotides | GAPDH | Qiagen | QT001199633 | |
| Peptide, recombinant protein | Recombinant murine leptin | Tocris, | Cat# TO-2985/1 | |
| Commercial assay or kit | Mouse leptin ELISA kit | BioVendor R and DR | Cat# RD291001200 | |
| Chemical compound, drug | 1,2,3,4-Tetrahydro-6 -nitro-2,3-dioxo-benzo [f]quinoxaline-7-sulfonamide (NBQX) | NIMH Chemical Synthesis and Drug Supply Program, https://nimh-repository.rti.org/ | | |

*Continued on next page*

*Continued*

| Reagent type (species) or resource | Designation | Source or reference | Identifiers | Additional information |
|---|---|---|---|---|
| Chemical compound, drug | D-2-amino-5-phosphovaleric acid (D-APV) | NIMH Chemical Synthesis and Drug Supply Program, https://nimh-repository.rti.org/ | | |
| Chemical compound, drug | 1,2,3,4-Tetrahydro-6-nitro-2,3-dioxo-benzo[f]quinoxaline-7-sulfonamide (NBQX) | NIMH Chemical Synthesis and Drug Supply Program, https://nimh-repository.rti.org/ | | |
| Chemical compound, drug | Isoguvacine | Tocris | Cat# 0235 | |
| Chemical compound, drug | VU0463271 | Tocris | Cat# 4719 | |
| Chemical compound, drug | Bumetanide | Sigma | Cat# B3023 | |
| Software and Algorithms | National Institutes of Health | | RRID:SCR_003070 | |
| Software and Algorithms | Synaptosoft, http://www.synaptosoft.com/MiniAnalysis/DownloadDemo.html | | | |

## Animals

All animal procedures were carried out in accordance with the European Union Directive of 22 September (2010/63/EU). Experiments were performed on both male and female postnatal day (P) 4 and 5 Wistar rats and P1 to P20 C57BL/6 transgenic mice lacking leptin (ob/ob) or leptin receptor (db/db) expression (purchased from Charles River Laboratory, Italy). Control experiments were performed on wild type (wt) littermates. Animals were housed in a temperature-controlled environment with a 12 light/dark cycle and free access to food and water. Mice were genotyped following the Jackson Laboratory genotyping protocol (strains B6.Cg-Lep[ob]/J, ID 000632 and B6.BKS-Lepr[db], ID 000697). For each experiment, tissues from littermate wt and KO mice were prepared and the persons assessing and quantifying the outcomes were blinded to the genotype of the mice.

To assess the effect of maternal obesity, 8 weeks old C57BL/6 females were fed with a high-fat diet (HFD, 60% kcal from fat, D12492, Research Diet) or a normal diet (ND, 10% kcal from fat, D12450B, Research Diet) during 6 weeks at the end of which they were weighed and mated. All females were maintained during gestation and lactation on the same diet received before.

## Leptin injection

Recombinant murine leptin was reconstituted in PBS buffer pH 7.4, and injected (5 mg/kg) sub-cutaneous in wt and *ob/ob* pups twice a day at 9–10 hr a.m and 5–6 h p.m. Control received same volume injections of vehicle, pH 7.4.

## Leptin immunoassay

The plasma samples were centrifuged (10.000 rpm, 10 min, 4°C) immediately after collection of arteriovenous blood samples obtained from 0 to 20 days old wt and *db/db* mice at 10–11 hr a.m. Plasma samples from leptin-treated mice (*ob/ob* and wt mice) were obtained 30 min to 1 hr after the last sub-cutaneous injection. Plasma was collected and stored at −80°C. Quantification of endogenous leptin was performed with Mouse Leptin ELISA Kit (BioVendor R and D) in the concentrated solutions following the manufacturer's protocol. The measured concentration of samples was calculated from the standard curve and expressed as ng/ml.

## Hippocampal slice preparation and electrophysiological recordings

Brains were removed and immersed into ice-cold (2–4°C) artificial cerebrospinal fluid (ACSF) with the following composition (in mM): 126 NaCl, 3.5 KCl, 2 $CaCl_2$, 1.3 $MgCl_2$, 1.2 $NaH_2PO_4$, 25 $NaHCO_3$ and 11 glucose, pH 7.4 equilibrated with 95% $O_2$ and 5% $CO_2$. Hippocampal slices (600 μm thick) were cut with a McIlwain tissue chopper (Campden Instruments Ltd.) and kept in ACSF at room temperature (25°C) for at least one hour before recording. Slices were then transferred to a submerged recording chamber perfused with oxygenated (95% $O_2$ and 5% $CO_2$) ACSF (3 ml/min) at 34°C.

$E_{GABA}$ measurement: Perforated patch-clamp recordings were made from CA3 pyramidal neurons using an axopatch 200B (Axon Instrument) or Multiclamp 700B (Molecular devices) amplifier. Glass recording electrodes had resistances of 4–7 MΩ when filled with KCl solution containing 150 mM KCl and 10 mM HEPES, pH adjusted to 7.2 with Tris-OH. The pipettes were tip filled with a gramicidin-free KCl solution and then backfilled with the same solution containing gramicidin A (50 μg/ml, diluted from a 50 mg/ml stock solution in DMSO). After the access resistance had dropped (40 to 80 MΩ) and stabilized (15–30 min), a current-voltage relationship was constructed by measuring the peak amplitude of averaged evoked GABAergic synaptic current (3 single sweeps) at different holding potentials in 10 mV increment recorded in the presence of glutamatergic receptor antagonists (NBQX 5 μM and D-APV 40 μM). Measurements were not corrected for the liquid junction potentials. A linear regression was used to calculate the best-fit line of the voltage dependence of the synaptic currents. Spontaneous rupture into whole-cell was evidenced by large inward synaptic currents due to $E_{Cl}$ of 0 mV.

Isoguvacine effect on neuronal firing: Loose cell attached patch clamp recordings were performed from CA3 pyramidal neurons using an axopatch 200B (Axon Instrument) with glass electrodes (4–7 MΩ) filled with KCl solution containing 150 mM KCl and 10 mM HEPES, pH adjusted to 7.2 with Tris-OH. After a baseline period of at least 10 min in the presence of NBQX (5 μM) and D-APV (40 μM), isoguvacine (10 μM) was bath applied for 2 min. The effect of isoguvacine was quantified as the mean frequency of action potential following application of isoguvacine (4–8 min) versus baseline frequency (−10–0 min). Synaptic activity was recorded with Axoscope software version 8.1 (Axon Instruments) and analyzed offline with Mini Analysis Program version 6.0 (Synaptosoft).

## Western blotting

Whole hippocampi were obtained from P5-6 mice as described above. Hippocampi were homogenized in RIPA buffer (150 mM NaCl, 1% Triton X-100, 0.1% SDS, 50 mM Tris HCl, pH 8, containing proteinases and phosphatases inhibitors (Complete Mini, Roche). Lysates were centrifuged (1.000 g for 10 min at 4°C) and the supernatant was heated at 90°C for 5 min with Laemmli loading buffer. Loading was 20 μg of proteins as determined using a BCA protein Assay Kit (Thermo Scientific). Proteins were separated in 7–15% SDS-PAGE and electrophoretically transferred to nitrocellulose membranes. Membranes were blocked with 5% bovine serum albumin (BSA) in TBS 0.1% Tween 20 (TBST) for 2 hr at RT, then incubated with primary antibodies diluted in TBST containing 3% BSA overnight at 4°C or 2 hr at RT. Blots were probed with antibody against KCC2 (1:2000; rabbit, US Biological) and tubulin (1:10.000; ß-tubulin, mouse, Sigma). After washing with TBST, membranes were incubated with HRP-conjugated secondary antibodies diluted in TBST containing 3% BSA for 60 min, washed with TBST, and then developed using the G:BOX gel imaging system (Syngene). Expression levels were estimated by ImageJ software.

## Real-time qRT-PCR

Whole hippocampi were obtained from P6 mice as described above. RNA was isolated and quantified by reading the absorbance at 260 nm (NanoPhotometer, IMPLEN) using Mini RNeasy kit (Qiagen), then converted to cDNA using 1 μg RNA and a QuantiTect Reverse Transcription kit (Qiagen) according to manufacturer's instructions. PCR was carried out with the LightCycler 480 SYBR Green I Master (Roche Applied Science) with 1 μL cDNA using the following oligonucleotides (QuantiTect Primer Assays, Qiagen): NKCC1 (Slc12a2; QT00197785), KCC2 (Slc12a5; QT00145327) and glyceraldehyde-3-phosphate dehydrogenase (GAPDH, QT001199633). Quantitative RT-PCR was performed with a Roche LC480 Light Cycler (Roche Applied Science) following the manufacturer's instructions. Relative mRNA values were calculated using the LC480 software and GAPDH as the housekeeping gene. PCR was performed in replicate of 3.

## Immuno-staining

Brains were removed from P6 mice and fixed overnight at 4°C in 4% paraformaldehyde (PFA). Brains were rinsed in phosphate buffer saline (PBS, 0.1M) and coronal sections (70 µM thick) were obtained using a vibratome (Microm HVM 650V). Section were incubated first for 1 hr in PBS with 1% bovine serum albumin (BSA) and 0.3% Triton X-100, then overnight at 4°C with a rabbit anti-panKCC2 primary antibody (1:4000; non commercial, gift from Dr. C. Rivera). Sections were rinsed in PBS and incubated for 2 hr with an Alexa Fluor 488 donkey anti-rabbit secondary antibody (1:1000, FluoProbes). Sections were counterstained for Nissl bodies, rinsed in PBS and mounted on microscope slides using Vectashield mounting medium (Vector). Immuno-reactivity was visualized using a laser scanning confocal microscope (LSM 510 Meta, Zeiss) with a 20X air objective and a 63X oil immersion objective. Optical sections were digitized (1024 × 1024 pixels) and processed using ImageJ software (National Institutes of Health, http://rsb.info.nih.gov/ij/). Analysis of the intensity of the distribution of KCC2 fluorescence was performed at high magnification (×63 objective) using the Image J program. The same straight line length (3 µm) was applied from the nucleus to the external cell compartment. The Plot profile values were analyzed and we normalized the fluorescence intensity to the highest intensity of the control condition. The intensity of KCC2 staining in neuronal cells was expressed as the mean ratio of KCC2/Neurotracer staining intensity.

## Primary cultures, transfection of rat hippocampal neurons, electrophysiological recordings, live cell immuno-labelling, immuno-staining and immunoprecipitation with phosphorylation site–specific antibodies

Hippocampi from 18-day-old rat embryos were dissected and dissociated using trypsin (0.05%) and plated at a density of 70,000 cells cm−2 in minimal essential medium (MEM) supplemented with 10% Nu-Serum (BD Biosciences), 0.45% glucose, 1 mM sodium pyruvate, 2 mM glutamine, and penicillin-streptomycin (10 IU ml−1) as previously described (*He et al., 2014*). On days 7, 10, and 13 of culture incubation (DIV, days in vitro), half of the medium was changed to MEM with 2% B27 supplement (Invitrogen). For electrophysiology neuronal cultures were plated on coverslips placed in 35 mm culture dishes. Twenty-four hours before plating, dishes with coverslips were coated with polyethylenimine (5 mg/ml).

For the analysis of KCC2 immuno-staining, KCC2-GFP clusters and electrophysiological experiments, pyramidal neurons were selected based on their morphology (*Benson et al., 1994*).

Gramicidin-perforated patch-clamp recordings were performed at 23–24°C as previously reported (*Friedel et al., 2015*) (details are in supporting information). Series resistance was monitored routinely at a Vh of −80 mV with 5 mV hyperpolarizing pulses, typically taking 10 to 15 min for the series resistance to stabilize at 15 to 60 MΩ). Data were low pass–filtered at 2 kHz and acquired at 10 kHz. Isoguvacine (30 mM) was focally applied (50 to 150 ms, 10,000 to 30,000 Pa) to the neuron soma and proximal dendrites through a micropipette connected to a Picospritzer (General Valve Corporation). Isoguvacine responses were recorded at voltages −120,−100, −80, and −60 mV. A linear regression was used to calculate the best-fit line of the voltage dependence of the synaptic currents. For analysis of Cl- extrusion kinetic all neurons were overloaded with identical amount of [Cl-]$_i$ resulting in E$_{GABA}$=-40 mV (~30 mM of [Cl-]$_i$). This was achieved by repetitive three-pulse applications of isoguvacine (100 ms pulses with interval 500 ms) at 0 mV followed by a single isoguvacine application (100 ms) at −40 mV until disappearance of the outwardly directed isoguvacine-induced current at −40 mV. This later corresponded to E$_{GABA}$=-40 mV. 20 s after the loading protocol, a pair of 100 ms isoguvacine pulses at −40 mV and −80 mV (interval 500 ms) was applied every 10 s to monitor the E$_{GABA}$. For quantification of Cl- extrusion kinetic, E$_{GABA}$ at each time point was first normalized to the mean control, pre-loading E$_{GABA}$ values. These normalized values were then normalized to the normalized E$_{GABA}$ obtained 20 s after the loading protocol (−49 ± 2 mV for control experiments, n = 7 neurons, 3 cultures).

For transfection of cultures growing in 35 mm dishes, 300 ml of Opti-MEM was mixed with 7 ml of *Ahima and Flier, 2000* (Invitrogen), 1 ml of Magnetofection CombiMag (OZ Biosciences), and 1 to 1.5 mg of different pcDNAs premixed in desired proportions. The mixture was incubated for 20 min at room temperature and thereafter distributed dropwise above the neuronal culture. Culture dishes were placed on a magnetic plate (OZ Biosciences) and incubated for 30 to 35 min at 37°C.

Transfection was terminated by the substitution of 90% of the incubation solution with fresh culture medium. The experiments were based on cotransfection into the same cell of two different pcDNAs encoding a fluorescent marker of transfection (eGFP, 0.3 μg), shRNAs (1.2 μg) against leptin receptor or WNK1, scrambled shRNAs or WNK1 construct. We used two shRNAs targeting different regions on the LepRb gene. The efficacy of these shRNAs has been tested in a previous study on human kidney cells and cultured hippocampal neurons (*Dhar et al., 2014*). The efficacy of the WNK1 construct and shRNA has been tested in a previous study in PC-12 and N2a cell line and cultured hippocampal neurons (*Friedel et al., 2015*).

For immunocytochemistry on living neurons, rabbit anti-GFP antibody was diluted in culture medium and applied to neurons for 2 hr at 37°C, 5% CO2. Neurons were then rinsed three times for 10 min at room temperature with Hepes-buffered saline solution containing 150 mM NaCl, 2.5 mM KCl, 2.0 mM MgCl2, 2.0 mM CaCl2, 20 mM Hepes, and 10 mM D-glucose (pH 7.4), labeled with anti-rabbit Cy3-conjugated antibody (dissolved in the Hepes buffered saline) for 20 min at 13°C and fixed in Antigenfix (Diapath). To reveal intracellular pool of live-labelled proteins, cells were subsequently permeabilized with 0.3% Triton X-100, blocked by 5% goat serum and incubated during 1 hr at room temperature (RT) with anti-rabbit Alexa 647-conjugated antibody. Then, for visualization of the entire pool of overexpressed KCC2-pH$_{ext}$ cells were labeled overnight (4°C) with mouse anti-GFP antibody and for 1 hr at RT with anti-mouse Alexa 488-conjugated antibody. For control of the cell integrity during live-cell immunolabelling, one batch of cultures were routinely transfected with KCC2 construct harboring phluorine tag linked to the intracellular N-terminus of the transporter.

Images of labeled cells were acquired with an Olympus Fluorview-500 confocal microscope (oil-immersion objectives 40x, (NA1.0) or 60x (NA1.4); zoom 1–5). We randomly selected and focused on a transfected cell by only visualizing Alexa-488 fluorescence and then acquired Z-stack images of Alexa-488, CY3 and Alexa-647 fluorochromes emitted fluorescence using, respectively green (excitation 488 nm, emission 505–525 nm), red (excitation 543 nm emission 560–600 nm) and infra-red (excitation 633, emission >660 nm) channels of the microscope. Each Z-stack included 10 planes of 1 μm optical thickness and taken at 0.5 μm distance between planes. The cluster properties and fluorescence intensities of each cell were analyzed with Metamorph software. First, we used the logical 'NOT' conversion of pairs of Alexa-647 and CY3 images to isolate in each focal plane the Alexa-647 signal that was not overlapping with CY3 fluorescence restricted to plasma membrane. This gave rise to additional images reflecting the fluorescence of the internalized pool of labeled clusters, called thereafter 'NOT-conversion'. Second, the arithmetic summation for each Z-stack and channel was performed to collect the whole fluorescence of the different signals (Alexa-488 = total protein fluorescence; CY3 = plasma membrane restricted fluorescence; NOT-conversion = internalized restricted fluorescence; Alexa-647 = all surface labeled fluorescence). Third, a binary mask was created for each cell from Alexa-488 image to isolate the signal coming from the transfected neuron, and the fluorescence parameters (total fluorescence, single cluster fluorescence as well as density and brightness of clusters) were analyzed for each channel (Alexa-488, CY3, NOT-conversion and Alexa-647) in regions overlapping with the binary mask. The analysis parameters were the same for each experiment and all experiments were done blind. After analysis, data were normalized to the mean value of cells transfected with KCC2-pHext + scrambled shRNA.

For immuno-staining, hippocampal cultures (75,000 cells/cm (*An and Xiang, 2016*), DIV 14) were fixed in 4% PFA-sucrose for 10 min. Coverslips were washed in PBS for 15 min and incubated in 0.2 M glycine for 10 min. Blocking was done in 1% BSA/0.5% Triton X-100 in PBS for 30 min to permeabilize cells and reduce nonspecific binding. Cultures were washed and incubated with a rabbit anti-panKCC2 (1:4000; non commercial, gift from Dr. C. Rivera) coupled to chicken anti-MAP2 (1:2000, Microtubule-associated protein 2, Sigma) antibodies in PBS overnight at 4°C. Primary antibodies were visualized after staining with the appropriate goat anti-rabbit and anti-donkey IgG conjugated to Cy3 (1:1000, chemicon) and Alexa488 (1:1000, FluoProbes), respectively in 1% BSA/PBS for 60 min. Cultures were washed and coverslips mounted using Vectashield (Vector). Sequential acquisition of immunoreactivity was visualized using laser scanning confocal microscope (Zeiss LSM 510 Meta) with a x63 oil-immersion objective. In each image, laser light levels and detector gain and offset were adjusted to avoid any saturated levels. Confocal micrographs are digital composites of a Z-series scan of 4–6 optical sections through a depth of 4–5 μm.

For the immunoprecipitation with phosphorylation site–specific antibodies, KCCs phosphorylated at the KCC2 Thr906 and Thr1007 equivalent residue were immuno-precipitated from clarified control

and leptin (100 nM, 24 hr) treated-hippocampal culture lysates (centrifuged at 16,000 g at 4°C for 20 min) using phosphorylation site–specific antibody coupled to protein G–Sepharose. The phosphorylation site–specific antibody was coupled with protein G–Sepharose at a ratio of 1 mg of antibody per 1 ml of beads in the presence of lysate (20 mg/ml) to which the corresponding nonphosphorylated peptide had been added. Two milligrams of clarified cell lysate was incubated with 15 mg of antibody conjugated to 15 ml of protein G–Sepharose for 2 hr at 4°C with gentle agitation. Beads were washed three times with 1 ml of lysis buffer containing 0.15 M NaCl and twice with 1 ml of buffer A. Bound proteins were eluted with 1 × LDS sample buffer.

Cell lysates (15 mg) in SDS sample buffer were subjected to electrophoresis on polyacrylamide gels and transferred onto nitrocellulose membranes. The membranes were incubated for 30 min with TTBS containing 5% (w/v) skim milk. The membranes were then immunoblotted in 5% (w/v) skim milk in TTBS with the indicated primary antibodies overnight at 4°C. Antibodies prepared in sheep were used at a concentration of 1 to 2 mg/ml. The incubation with phosphorylation site–specific sheep antibodies was performed with the addition of the non-phosphorylated peptide antigen (10 mg/ml) used to raise the antibody. The blots were then washed six times with TTBS and incubated for 1 hr at room temperature with secondary HRP-conjugated antibodies diluted 5000-fold in 5% (w/v) skim milk in TTBS. After repeating the washing steps, the signal was detected with the enhanced chemiluminescence reagent. Imumunoblots were developed by ChemiDoc Imaging Systems (Bio-Rad). Figures were generated using Photoshop and Illustrator (Adobe). The relative intensities of immunoblot bands were determined by densitometry with ImageJ software.

Antibodies used for western blots were raised in sheep and affinity-purified on the appropriate antigen by the Division of Signal Transduction Therapy Unit (DSTT) at the University of Dundee; other antibodies were purchased. NKCC1 total (S022D, first bleed, raised against residues 1–288 of human NKCC1); KCC2a phosphoT906 (S959C, first bleed; raised against residues 975–989 of human KCC3a phosphorylated at T991, SAYTYER(T)LMMEQRSRR); KCC2a phosphoT1007 (S961C, first bleed; raised against residues 1032–1046 or 1041–1055 of human KCC3a phosphorylated at T1048). KCC2 Ser940 antibody (NBP2-29513, Novus Biologicals). KCC2 total antibody (S700C, first bleed; raised against residues 1–119 of human KCC2A); The anti-β-Tubulin III (neuronal) antibody (T8578) was purchased from Sigma-Aldrich. Secondary antibodies coupled to horseradish peroxidase used for immunoblotting were obtained from Pierce. IgG used in control immunoprecipitation experiments was affinity-purified from pre-immune serum using Protein G-Sepharose.

## Reagents

The following reagents were purchased from the indicated sources: 1,2,3,4-Tetrahydro-6-nitro-2,3-dioxo-benzo[f]quinoxaline-7-sulfonamide (NBQX) and D-2-amino-5-phosphovaleric acid (D-APV) from the Molecular, Cellular, and Genomic Neuroscience Research Branch (MCGNRB) of the National Institute of Mental Health (NIMH, Bethesda, MD, USA). Leptin, Isoguvacine and VU0463271 from Tocris Cookson (Bristol, UK). Bumetanide from Sigma (St Louis, MO, USA).

## Statistics

No statistical methods were used to predetermine sample sizes, but our sample sizes correspond to those reported in previous publications (Tyzio et al., 2006; Dhar et al., 2014; Friedel et al., 2015). To ensure the consistency and reproducibility of our results, we conducted repeated trials in different cell cultures, acute brain slice and hippocampi prepared from at least three different animals for each experimental condition. In this study, the persons performing experiments and analyzing the data were blinded to the genotype of the mice. These include electrophysiological recordings on acute slices, western blot, and PCR. The one way ANOVA followed by a Tukey's *post hoc* test was used for multiple comparisons between experimental conditions. A two-tailed unpaired Student's *t*-test was used to analyze difference between two individual groups. A two-tailed paired Student's *t*-test was used to analyze differences within one group across conditions, that is frequency of action potential before and after isoguvacine. All data are expressed as Mean ±standard error to the mean (S.E.M.). Data are judged significantly different when $p < 0.05$. Statistical information is provided in the figures, figure legends and text.

## Acknowledgments
We thank Dr. Y Ben-Ari for helpful support and suggestions during this study; Drs. R Cossart and NKourdougli for reading of the manuscript and Drs. E Cherubini, C Rivera and E Sernagor for helpful critical comments on the study. This work was supported by the Ministère de la Recherche et de l'Enseignement Supérieur, Neurochlore (CD) and the National Institutes of Health (Grant MH086032, GW).

## Additional information

### Funding

| Funder | Grant reference number | Author |
| --- | --- | --- |
| National Institutes of Health | MH086032 | Gary Wayman |
| National Institutes of Health | 1RO1HD092396 | Gary Wayman Jean-Luc Gaiarsa |

The funders had no role in study design, data collection and interpretation, or the decision to submit the work for publication.

### Author contributions
Camille Dumon, Jean-Luc Gaiarsa, Writing—original draft, Writing—review and editing, Performed electrophysiological experiments on slices, western blot experiments and analysed the data; Diabe Diabira, Jean-Luc Gaiarsa, Performed electrophysiological experiments on slices and analyzed data; Ilona Chudotvorova, Prepared the neuronal cultures and performed the multi-step immunostaining; Francesca Bader, Bred the colonies; Semra Sahin, Performed western blots experiments on neuronal cultures and analyzed the data; Jinwei Zhang, Performed the immunoprecipitation experiments and the western blot on neuronal cultures and analyzed the data; Christophe Porcher, Performed qPCR and immunostaining experiments and analysed the data; Gary Wayman, Writing—original draft; Igor Medina, Performed electrophysiological experiments on neuronal cultures and analyzed data

### Author ORCIDs
Camille Dumon https://orcid.org/0000-0002-1625-3091
Jinwei Zhang https://orcid.org/0000-0001-8683-509X
Igor Medina https://orcid.org/0000-0001-6839-5414
Jean-Luc Gaiarsa https://orcid.org/0000-0001-7354-0559

### Ethics
Animal experimentation: All animal procedures were carried out in accordance with the European Union Directive of 22 September (2010/63/EU).

### Decision letter and Author response
Decision letter https://doi.org/10.7554/eLife.36726.020
Author response https://doi.org/10.7554/eLife.36726.021

## Additional files

### Supplementary files
• Supplementary file 1. Advanced onset of GABA inhibition in leptin deficient mice.
DOI: https://doi.org/10.7554/eLife.36726.017
• Transparent reporting form
DOI: https://doi.org/10.7554/eLife.36726.018

## Data availability

All data generated or analyszed during this study are included in the manuscript and supporting files. Source data files have been provided for Figures 1 to 4 and Supplementary Figures 1 and 2.

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
