## [Decision Letter]

Thank you for submitting your article "The adipocyte hormone leptin sets the emergence of hippocampal inhibition in mice" for consideration by *eLife*. Your article has been reviewed by three peer reviewers, one of whom is a member of our Board of Reviewing Editors, and the evaluation has been overseen by Gary Westbrook as the Senior Editor. The reviewers have opted to remain anonymous. The reviewers have discussed the reviews with one another and the Reviewing Editor has drafted this decision to help you prepare a revised submission.

Summary:

The work by Dumon et al. investigates the role of the adipolytic hormone Leptin. The loss of leptin or its receptor delays the maturation of GABAergic inhibition and maternal obesity, which is associated with hyperleptinemia, delays the switch from depolarizing to hyperpolarizing inhibition. The study examines the effect of leptin on the KCC2 transporter and finds that leptin shifts the reversal potential of GABA to more positive values and delays the extrusion of chloride in cultured cells preloaded with chloride. Using an immunolabeling protocol, the authors show that leptin treatment causes a decline in the expression of the KCC2 transporter inserted in the neurons' membrane but increases the amount of the internalized transporter in the cells' cytoplasm, suggesting a reduced membrane bonding of KCC2 in cultured cells. Applying a maternal obesity model, the authors show that hyperleptinemia during maternal obesity delays the developmental switch of GABA reversal potential and down regulates KCC2 in the offspring. It is a technically well-performed study with many controls and several genetically modified mouse lines. The data are very convincing and of interest to a broad readership. Some aspects were, however, not clear as outlined below.

Essential revisions:

1) The authors make the point that phosphorylation of the C-terminus in particular the threonine 906 and 1007 residue inhibits KCC2 activity and enhances its endocytosis. The authors therefore assed the phosphorylation state of KCC2. They mention in the text that the amount of phosphorylated KCC2 increased after leptin treatment. The Western plot shown in Figure 2C, however, indicates the opposite. Moreover, based on the statement of the authors relative expressions of phospho-S940 were unchanged, however, Figure 2C again indicates a reduction. Please clarify.

2) One of the reviewers stated that experiments shown in Figures 1 and 2 do not contribute in any way to clarify if leptin (and its receptor) are contributing to the timing of the GABA shift, but they just indicate that leptin is capable to modulate E_GABA_ (and KCC2) in more mature neurons in which the GABA shift has already occurred. Therefore, we suggest to move the figures to the second part of the manuscript or to the supplement.

3) Next, the authors tested whether WNKI activity is required for the KCC2 T906 and T 1007 phosphorylation. By using sh-RNA for downregulation of WNK1 or by using a constitutively active WNK1 mutant, the authors show that indeed WNK1 is required for the depolarizing shift induced by leptin treatment. Here, it would be important to examine by using Western blot analysis, whether the expression of the NKCC2 transporter was altered and, thus, may influence the measurement of the chloride reversal potential. This criticism applies also to the data shown in Figure 3.

4) All experiments shown in Figures 1 and 2 are based on cultured cells. Please provide information on the nature of the recorded cells (glutamatergic / GABAergic), if possible.

5) The authors should suggest what behaviors and physiologic parameters might be altered secondary to increased or decreased leptin during perinatal overnutrition.

6) In the Abstract, the factors controlling the GABA shift are mentioned as "largely unknown". On the contrary, several factors have been shown to be involved in the regulation of the GABA shift. The author should reduce their claim here, as leptin is just one of the many factors that "sets the tempo for the emergence of the GABAergic inhibition" in the hippocampus.

7) The authors should show if and how leptin modifies E_GABA_ and KCC2 expression during neuronal maturation in culture at different DIV. This is an important point because effects and signaling mechanisms of factors modulating KCC2 and the GABA shift could change during the maturation of hippocampal neurons in culture (Leonzino 2016).

8) Quantification of NKCC1 should also be performed to check for cross regulation between the two cotransporters.

9) An alternative method for KCC2 quantification should be applied. One possibility is to perform biotynilation of membrane proteins followed by immunoprecipitation.

---

## [Author Response]

Essential revisions:1) The authors make the point that phosphorylation of the C-terminus in particular the threonine 906 and 1007 residue inhibits KCC2 activity and enhances its endocytosis. The authors therefore assed the phosphorylation state of KCC2. They mention in the text that the amount of phosphorylated KCC2 increased after leptin treatment. The Western plot shown in Figure 2C, however, indicates the opposite. Moreover, based on the statement of the authors relative expressions of phospho-S940 were unchanged, however, Figure 2C again indicates a reduction. Please clarify.

We agree with the reviewer and have amended the manuscript to clarify this point.

The Western blots do show a decrease in the amount of the different phosphorylated forms of KCC2 (threonine 906 and 1007, serine 940) following the treatment with leptin. However, the amount of total KCC2 was also strongly diminished. In the manuscript we therefore referred to the relative amount of the phosphorylated forms of KCC2 defined as the ratio of the phosphorylated forms of KCC2 to total KCC2. We found that although the amount of phospho-T 906 and 1007 was decreased, their relative amount was increased after leptin treatment. The relative amount of phospho-S940 was unchanged. We have amended the revised manuscript to clarify this point (subsection “Leptin controls chloride homeostasis in vitro”, fourth paragraph).

2) One of the reviewers stated that experiments shown in Figures 1 and 2 do not contribute in any way to clarify if leptin (and its receptor) are contributing to the timing of the GABA shift, but they just indicate that leptin is capable to modulate E_GABA_ (and KCC2) in more mature neurons in which the GABA shift has already occurred. Therefore, we suggest to move the figures to the second part of the manuscript or to the supplement.

We agree with the reviewer. We have moved the Figures 1 and 2, and the corresponding text to the second part of the revised manuscript.

3) Next, the authors tested whether WNKI activity is required for the KCC2 T906 and T 1007 phosphorylation. By using sh-RNA for downregulation of WNK1 or by using a constitutively active WNK1 mutant, the authors show that indeed WNK1 is required for the depolarizing shift induced by leptin treatment. Here, it would be important to examine by using Western blot analysis, whether the expression of the NKCC2 transporter was altered and, thus, may influence the measurement of the chloride reversal potential. This criticism applies also to the data shown in Figure 3.

We agree with the reviewer. We provide additional results on the expression of NKCC1 in *db/db* mice and leptin-treated neuronal cultures.

Western blot analysis revealed that NKCC1 expression was not different in *db/db* mice and wt mice (subsection “Advanced onset of GABAergic inhibition in leptin-receptor deficient mice”, last paragraph).

Regarding the neuronal cultures, it is not possible to assess the expression of NKCC1 by Western blot analysis in transfected neurons due to the low percentage of neurons transfected per dish. We therefore assessed the expression of NKCC1 in non-transfected neuronal cultures treated with leptin (100 nM, 24h) and found that NKCC1 expression was decreased in leptin-treated cultures (subsection “Leptin controls chloride homeostasis in vitro”, second paragraph). How leptin down-regulates NKCC1 expression is presently unknown, but we found that the KCC2 to NKCC1 ratio was decreased, thus supporting the idea that chloride extrusion is reduced following leptin-treatment. Moreover, it should be noted that the measurement of E_GABA_ in the neuronal cultures were performed in the presence of bumetanide 10µM to exclude any influence of NKCC1.

4) All experiments shown in Figures 1 and 2 are based on cultured cells. Please provide information on the nature of the recorded cells (glutamatergic / GABAergic), if possible.

We have not done any specific experiments to characterize the nature of the recorded cells. However, based on morphological criteria and our previous experience (Chudotvorova et al., 2005), we believe that the neurons used for the analysis of KCC2-GFP clusters and for the electrophysiological experiments were glutamatergic cells. According to the study of Benson et al., 1994, most glutamatergic cells are spiny neurons with a pyramidal shaped soma, while most GABAergic cells are non-spiny and have more fusiform or polygonal shaped soma. The dendritic arbor is another criterion with glutamatergic neurons having more primary dendrites and shorter dendritic segments. This point is addressed in the revised manuscript (Materials and methods, subsection “Primary cultures, transfection of rat hippocampal neurons, electrophysiological recordings, live cell immuno-labelling, Immuno-staining and Immunoprecipitation with phosphorylation site–specific antibodies”).

5) The authors should suggest what behaviors and physiologic parameters might be altered secondary to increased or decreased leptin during perinatal overnutrition.

We agree with the reviewer and have amended the Discussion (fourth and fifth paragraphs). Human and rodents studies show that excess of leptin (maternal obesity, over nutrition) or deficit in leptin (maternal separation, under nutrition) increase the risk of developing mental and behavioral disorders, such as autism spectrum disorders, anxiety, depression and cognitive impairments.

6) In the Abstract, the factors controlling the GABA shift are mentioned as "largely unknown". On the contrary, several factors have been shown to be involved in the regulation of the GABA shift. The author should reduce their claim here, as leptin is just one of the many factors that "sets the tempo for the emergence of the GABAergic inhibition" in the hippocampus.

We agree with the reviewer. As stated in our manuscript, oxytocin and the thyroid hormone also control the GABAergic developmental sequence. We removed this sentence from the Abstract.

7) The authors should show if and how leptin modifies E_GABA_ and KCC2 expression during neuronal maturation in culture at different DIV. This is an important point because effects and signaling mechanisms of factors modulating KCC2 and the GABA shift could change during the maturation of hippocampal neurons in culture (Leonzino 2016).

We thank the referee for this suggestion. Indeed, if leptin down-regulates the expression of KCC2 by activating the WNK pathway, leptin can be expected to have no effect, or a moderate effect, in the immature neuron at a time when KCC2 expression is low and WNK activity is high as shown in our previous study (Friedel et al., 2015). In line with this assumption, we found no significant difference in our E_GABA_ measurement between control and leptin-treated immature (6 DIVs) neuronal cultures (subsection “Leptin controls chloride homeostasis in vitro”, first paragraph, Figure 2B).

8) Quantification of NKCC1 should also be performed to check for cross regulation between the two cotransporters.

We agree with the referee. As explained in the comment 3, we have assessed the expression of NKCC1 in *db/db* mice and leptin-treated neuronal cultures and have amended the manuscript accordingly. Western plot analysis revealed that NKCC1 expression was not different in *db/db* mice and wt mice (subsection “Advanced onset of GABAergic inhibition in leptin-receptor deficient mice”, last paragraph), while NKCC1 expression was reduced in leptin-treated neuronal cultures (subsection “Leptin controls chloride homeostasis in vitro”, second paragraph). Also, to exclude possible cross regulation between the two cotransporters, the measurements of E_GABA_ in neuronal cultures were performed in the presence of bumetanide (10µM), and the effect of leptin-treatment were occluded by VUO463271 (10µM), highlighting the contribution of KCC2.

9) An alternative method for KCC2 quantification should be applied. One possibility is to perform biotynilation of membrane proteins followed by immunoprecipitation.

As suggested by the reviewer, we tried to perform biotinylation experiments in neuronal cultures. We faced two problems. The first problem is the low relative amount of biotinylated KCC2. The second problem is biotin leakage, which results in the presence of a cytoplasmic protein (tubulin) in the biotinylated fraction. We are therefore not confident in the results we have obtained with this method. We therefore performed immuno-staining experiments as an alternative method to assess the effect of leptin on KCC2 expression. With this approach, we confirmed that KCC2 expression was up-regulated in db/db mice and down-regulated in leptin-treated hippocampal cultures (subsection “Advanced onset of GABAergic inhibition in leptin-receptor deficient mice”, last paragraph and subsection “Leptin controls chloride homeostasis in vitro”, second paragraph, Figure 1—figure supplement 2).